# Direct observation of exceptional points in coupled photonic-crystal lasers with asymmetric optical gains

Kyoung-Ho Kim[1,*], Min-Soo Hwang[1,*], Ha-Reem Kim[1], Jae-Hyuck Choi[1], You-Shin No[1] & Hong-Gyu Park[1]

Although counter-intuitive features have been observed in non-Hermitian optical systems based on micrometre-sized cavities, the achievement of a simplified but unambiguous approach to enable the efficient access of exceptional points (EPs) and the phase transition to desired lasing modes remains a challenge, particularly in wavelength-scale coupled cavities. Here, we demonstrate coupled photonic-crystal (PhC) nanolasers with asymmetric optical gains, and observe the phase transition of lasing modes at EPs through tuning of the area of graphene cover on one PhC cavity and systematic scanning photoluminescence measurements. As the gain contrast between the two identical PhC cavities exceeds the intercavity coupling, the phase transition occurs from the bonding/anti-bonding lasing modes to the single-amplifying lasing mode, which is confirmed by the experimental measurement of the mode images and the theoretical modelling of coupled cavities with asymmetric gains. In addition, we demonstrate active tuning of EPs by controlling the optical loss of graphene through electrical gating.

[1] Department of Physics, Korea University, Seoul 136-701, Republic of Korea. * These authors contributed equally to this work. Correspondence and requests for materials should be addressed to H.-G.P. (email: hgpark@korea.ac.kr).

The exceptional points (EPs), observed in the non-Hermitian parity–time (PT)-symmetric physical systems that possess multi-well potentials of energy gain or loss[1–14], have been recently explored in coupled optical cavities consisting of two identical micrometre-sized cavities, by controlling intercavity coupling and asymmetric gain or loss[15–19]. For example, EPs and unidirectional light transmission were measured in active and passive silica microtoroids coupled to tapered fibres, where the intercavity coupling was tuned[15,16]. In addition, several counter-intuitive lasing behaviours were observed in coupled cavities, including the suppression and enhancement of Raman lasing in coupled microtoroids with controlled asymmetric cavity loss[17], reversed pump dependence of lasing in coupled quantum-cascade microdisks under unbalanced electric pumping[18], and single-mode lasing in coupled microrings under asymmetric optical pumping[19]. On the other hand, photonic-crystal (PhC) cavities with high-quality factors and small mode volumes can also provide a useful platform for non-Hermitian wavelength-scale optical systems[20–23]. Only few lasing modes are supported by the PhC cavities with submicrometre-sized footprints, which allow clear mode identification and accurate analysis of lasing behaviours[20,24–27]. Furthermore, rational design and engineering of intercavity coupling in coupled PhC cavities have motivated the development of quantum–optical Josephson interferometry[28], spontaneous mirror-symmetry breaking[29] and self-pulsation lasers[30]. However, simplified and practical approaches, enabling the efficient access of EPs and the unambiguous phase transition of supermodes in wavelength-scale coupled cavities, have not been widely explored yet.

In this work, we demonstrate coupled PhC cavities with a monolayer graphene sheet that partially covers only one cavity to provide asymmetric optical gain and sufficient gain contrast. Scanning photoluminescence (PL) measurements show the bonding/anti-bonding lasing modes, single-amplifying lasing mode and the combination of these lasing modes are excited in the coupled cavities with no graphene, large-area and small-area graphene covers, respectively. These observed lasing modes are clearly identified using the measured mode images and the theoretical asymmetric gain model of non-Hermitian coupled cavities. In particular, the phase transition of lasing modes at EP can be directly measured with varying pump position and power in the coupled cavities with small-area graphene. Furthermore, optical loss of graphene is systematically controlled by electrical gating with ion gel for the demonstration of tunable EPs.

## Results

**Asymmetric gain model.** Figure 1a shows our coupled cavities that consist of two identical PhC cavities with three missing air holes, cavity 1 (yellow) and cavity 2 (blue). The complex eigenfrequencies in this system were calculated using a full-wave numerical simulation (symbols, Fig. 1b,c) with appropriate parameters, including the optical gains of cavities 1 and 2, $\gamma_1$ and $\gamma_2$, and the identical intrinsic optical loss in each cavity, $\kappa$. In the simulation, the optical gain was introduced only inside each cavity by systematically changing the extinction coefficient (see Methods)[18]. The real and imaginary parts of the calculated eigenfrequencies, Re($f$) and Im($f$), show unique features as $\gamma_1$ and $\gamma_2$ vary independently. Regarding Re($f$), when the two cavities have identical gains, $\gamma_1 = \gamma_2 = 1.106$ THz, the excited modes are split into two different supermodes (blue and red symbols, Fig. 1b) with a Re($f_\pm$) of 199.36 and 198.68 THz. This frequency splitting decreases in the region with $0.426 < \gamma_1 < 1.106$ and $\gamma_2 = 1.106$ THz, and increases in the region with $\gamma_1 = 0$ and

$0 < \gamma_2 < 0.680$ THz, whereas the supermodes coalesce in the region with $0 < \gamma_1 < 0.426$ and $\gamma_2 = 1.106$ THz or $\gamma_1 = 0$ and $0.680 < \gamma_2 < 1.106$ THz (Fig. 1b). In contrast, Im($f$) shows the opposite behaviour to Re($f$). The Im($f$) of both supermodes monotonically decreases exhibiting the same values, while bifurcating in the region where $0 < \gamma_1 < 0.426$ and $\gamma_2 = 1.106$ THz or $\gamma_1 = 0$ and $0.680 < \gamma_2 < 1.106$ THz (Fig. 1c). Both Re($f$) and Im($f$) are degenerate either at $\gamma_1 = 0.426$ and $\gamma_2 = 1.106$ THz or $\gamma_1 = 0$ and $\gamma_2 = 0.680$ THz, which are EPs.

The calculated eigenfrequencies agree well with the values obtained from the coupled mode theory in the non-Hermitian system of coupled optical cavities with asymmetric gains and identical losses (solid lines, Fig. 1b,c)[31,32]. In particular, the eigenfrequencies of supermodes in coupled cavities, $f_\pm$, are given by

$$f_\pm = f_0 \pm \sqrt{J^2 - (\Delta\gamma)^2} + i\left(\gamma_{\text{avg}} - \kappa\right) \qquad (1)$$

where $f_0$ is the eigenfrequency of a single cavity, $J$ is the coupling constant of coupled cavities, $\Delta\gamma = 1/2 \times |\gamma_2 - \gamma_1|$ is the gain contrast, and $\gamma_{\text{avg}} = 1/2 \times (\gamma_1 + \gamma_2)$ is the average gain (see Supplementary Note 1 for details). Notably, the coupled mode theory explains that the gain-dependent features of Re($f$) and Im($f$) of supermodes are attributed to the weighted relative difference between $\Delta\gamma$ and $J$. In our coupled cavities, $J$ is constant owing to the fixed numbers of PhC air holes between cavities 1 and 2, but $\Delta\gamma$ is varied as the asymmetric gain is applied. For example, for the identical gain ($\gamma_1 = \gamma_2 = 1.106$ THz or $\gamma_1 = \gamma_2 = 0$ THz), $\Delta\gamma$ is zero and the frequency splitting in Re($f_\pm$) is maximized. As the unbalanced gains of cavities 1 and 2 are provided, the relative difference between $\Delta\gamma$ and $J$ is reduced and the frequency splitting in Re($f_\pm$) becomes smaller. Eventually, the EPs occur at $\Delta\gamma = J$. If the gain contrast is greater than the coupling constant ($\Delta\gamma > J$), Re($f_\pm$) values are degenerate as $f_0$, whereas the degeneracy is broken in Im($f_\pm$), because all terms in Equation 1 become imaginary values except for $f_0$.

In addition, we calculated the $x$-component of the normalized electric fields of the supermodes, $E_x$, in the coupled PhC cavities when cavities 1 and 2 have identical gains ($\gamma_1 = \gamma_2 = 1.106$ THz; Fig. 1d,e) or asymmetric gains ($\gamma_1 = 0$ and $\gamma_2 = 1.106$ THz; Fig. 1f,g). The in-phase bonding mode (Fig. 1d) and the out-of-phase anti-bonding mode (Fig. 1e) exhibit identical intensities in both cavities. The amplifying mode is confined in cavity 2 (Fig. 1f), whereas the decaying mode is confined in cavity 1 (Fig. 1g). We note that Im($f$) determines whether the supermodes can be amplified (Im($f$) > 0) or dissipated (Im($f$) < 0)[18,31,32]. As shown in Fig. 1c, the bonding, anti-bonding and amplifying modes show Im($f$) > 0, whereas the decaying mode shows Im($f$) < 0: the stored energies of the bonding, anti-bonding and amplifying modes grow, but the energy of the decaying mode is dissipated. Consequently, the bonding and anti-bonding modes are observed as cavities 1 and 2 provide identical or small-contrast asymmetric gains ($\Delta\gamma < J$) or in the unbroken PT-symmetry phase, whereas a single-amplifying mode can be observed as the coupled cavities provide large-contrast asymmetric gains ($\Delta\gamma > J$) or in the broken PT-symmetry phase.

**Control of gain contrast by graphene.** To experimentally demonstrate distinctive lasing behaviours in the broken and unbroken PT-symmetry phases, we fabricated the coupled PhC cavities in the InGaAsP slab including a single quantum well (see Methods). As the two cavities were closely located with a centre-to-centre distance of $\sim$1.45 μm for proper intercavity coupling, it is very difficult to provide asymmetric optical gains and achieve sufficient gain contrast in such a wavelength-scale by

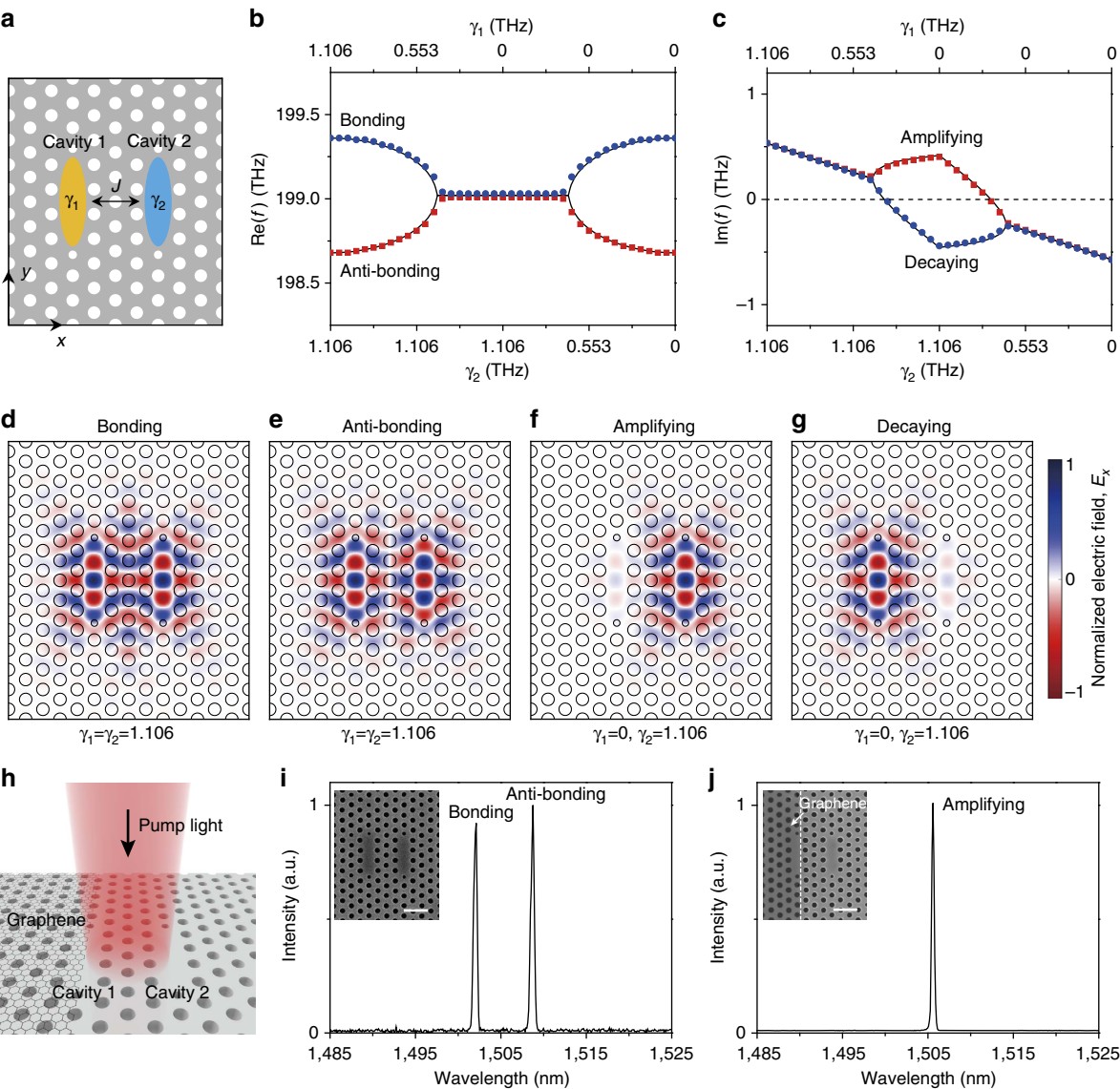

**Figure 1 | Asymmetric gain model and exceptional points in coupled PhC cavities.** (**a**) Schematic illustration of coupled PhC cavities that consist of two identical three-cell cavities in a triangular-lattice slab structure with a lattice constant of 420 nm, a regular hole diameter of 265 nm, and a reduced hole diameter of 140 nm. The slab thickness is 250 nm. Cavities 1 and 2 have optical gains, $\gamma_1$ and $\gamma_2$, and an intercavity coupling constant, $J$. In our model, $\gamma_1$ and $\gamma_2$ are provided only inside the cavities. (**b,c**) Real (**b**) and imaginary (**c**) parts of calculated complex eigenfrequencies using a full-wave simulation (blue and red symbols) and non-Hermitian coupled cavity model in Equation 1 (black solid lines), with independently varying $\gamma_1$ and $\gamma_2$. Blue and red symbols denote two different supermodes in the coupled PhC cavities. In the calculations, $J$ was fixed to 0.34 THz, while $0<\gamma_1<1.106$ THz and $0<\gamma_2<1.106$ THz. (**d–g**) Calculated $x$-components of normalized electric fields, $E_x$, for $\gamma_1=\gamma_2=1.106$ THz (**d** and **e**), and $\gamma_1=0$ and $\gamma_2=1.106$ THz (**f** and **g**). These modes were termed (**d**) bonding (Re($f$) = 199.36 THz), (**e**) anti-bonding (Re($f$) = 198.68 THz), (**f**) amplifying (Re($f$) = 199.02 THz) and (**g**) decaying (Re($f$) = 199.02 THz) modes. (**h**) Schematic illustration of the coupled PhC cavities with partially covered graphene. The centre of the coupled cavities is uniformly illuminated by a pumping laser. (**i**) Measured PL spectrum from the coupled cavities without graphene. The pumping power was 213 μW. Two lasing peaks were observed at wavelengths of 1502.1 and 1508.8 nm. Inset, fabricated coupled cavities without graphene. Scale bar, 1 μm. (**j**) Measured PL spectrum from the coupled cavities of panel **i** after introducing graphene cover. The pumping power was 303 μW. A single lasing peak was observed at a wavelength of 1505.6 nm. Inset, fabricated coupled cavities with graphene. The boundary of graphene cover is indicated by a white dashed line. Scale bar, 1 μm.

using conventional optical pumping methods[33,34]. To address this issue, we placed a monolayer graphene sheet on top of only one PhC cavity (Fig. 1h). By partially covering one cavity with atomic-scale thin graphene layer, the coupled cavities can experience a large gain contrast because of the broadband light absorption of graphene, with a negligibly small change in the intrinsic eigenfrequency of a single cavity mode[35–38]. For example, the gain of cavity 1 in Fig. 1h is effectively reduced by

the optical loss of the graphene: the effective gain of cavity 1 becomes $\gamma_{1,\text{eff}} = \gamma_1 - \kappa_{\text{graphene}}$, where $\kappa_{\text{graphene}}$ is the optical loss induced by the graphene in cavity 1 (Supplementary Note 1). In this model, $\kappa_{\text{graphene}}$ increases with enlarging the area covered by graphene, while $\gamma_2$ remains unaffected. The scanning electron microscopy (SEM) images of the fabricated coupled PhC cavities before and after applying the graphene cover on cavity 1 are shown in the insets of Fig. 1i,j, respectively. The

PL measurements were then carried out using a 980-nm pulsed pumping laser that illuminated both cavities (Fig. 1h). In the coupled PhC cavities without graphene, the bonding and anti-bonding lasing peaks were observed at wavelengths of 1502.1 and 1508.8 nm, respectively (Fig. 1i)[26,29]. However, after partially covering cavity 1 with the monolayer of graphene, a single lasing peak was observed at a wavelength of 1505.6 nm, which is located between the wavelengths of the bonding and anti-bonding lasing modes (Fig. 1j). These measurement results under symmetric optical pumping imply that the partial graphene cover not only decreases the optical gain of cavity 1 effectively but also enhances the gain contrast in the coupled cavities, and thus, the split supermodes have coalesced into the single lasing mode, as discussed in Fig. 1b (ref. 19).

**Measurements in the coupled cavities without graphene.** To clearly access the potential phase transition from the unbroken to broken PT-symmetry phases, we performed scanning PL measurements in the coupled PhC cavities with and without graphene (see Methods). A 980-nm pump laser with a spot size of $\sim 3.0\,\mu m$ and an incident peak pump power of $375\,\mu W$ was line-scanned with a scanning step of $0.2\,\mu m$ (Fig. 2a) and then the coupled cavities experienced asymmetric optical gains varying with pump position, $x_{pump}$ (Fig. 2b,c). First, we measured the PL spectra from the coupled PhC cavities without graphene (Fig. 2d). As the pump laser was line-scanned from $x_{pump} = -2.2$ (left end) to $2.2\,\mu m$ (right end), the bonding lasing peak (black line) at a wavelength of 1510.1 nm was observed in the entire pumping range, while the anti-bonding lasing peak (red line) at a wavelength of 1513.8 nm was only present in the pumping region of $-0.6 < x_{pump} < 0.6\,\mu m$. In Fig. 2e, we also measured the light in-light out curves (L–L curves) of the bonding (black line, at $x_{pump} = -0.6\,\mu m$) and anti-bonding modes (red line, at $x_{pump} = 0\,\mu m$) in the respective pump positions of maximum output intensities. The lasing thresholds of the bonding and anti-bonding modes were identical, $\sim 130\,\mu W$, although the above-threshold output intensity of the bonding mode was almost two times larger than that of the anti-bonding mode. In addition, we plotted the measured lasing peak intensities of Fig. 2d as a function of the pump position (Fig. 2f). The bonding lasing mode was dominantly excited, when the pump laser was well-aligned with cavity 1 ($x_{pump} = -0.6\,\mu m$) or cavity 2 ($x_{pump} = 0.6\,\mu m$), whereas the anti-bonding lasing mode was dominantly excited at the central pump position ($x_{pump} = 0\,\mu m$), owing to the different field overlap with the pump laser[29] and the mode competition between the two modes[39]. The insets of Fig. 2f clearly show lasing mode images of the bonding mode with a central intensity anti-node and the anti-bonding mode with a central node, as calculated in Fig. 1d,e (refs 26,29). Furthermore, the log-scale false-colour map of the measured PL spectra as a function of the pump position (Fig. 2g) reveals that the lasing wavelengths of the bonding and anti-bonding modes are almost constant with varying pump position. Consequently, these results indicate that the asymmetric optical gain was provided by position-dependent local pumping, but the gain contrast was not sufficiently increased in all pump positions compared with the intercavity coupling, as shown in the unbroken PT-symmetry phase ($\Delta\gamma < J$).

**Measurements in the coupled cavities with large-area graphene.** Next, we measured PL spectra from the coupled PhC cavities with graphene on top of cavity 1, while the pump laser with an incident peak power of $303\,\mu W$ was line-scanned in the same manner, as in Fig. 2 (see Methods). Two types of graphene layers were examined to study the effect of graphene loss on the

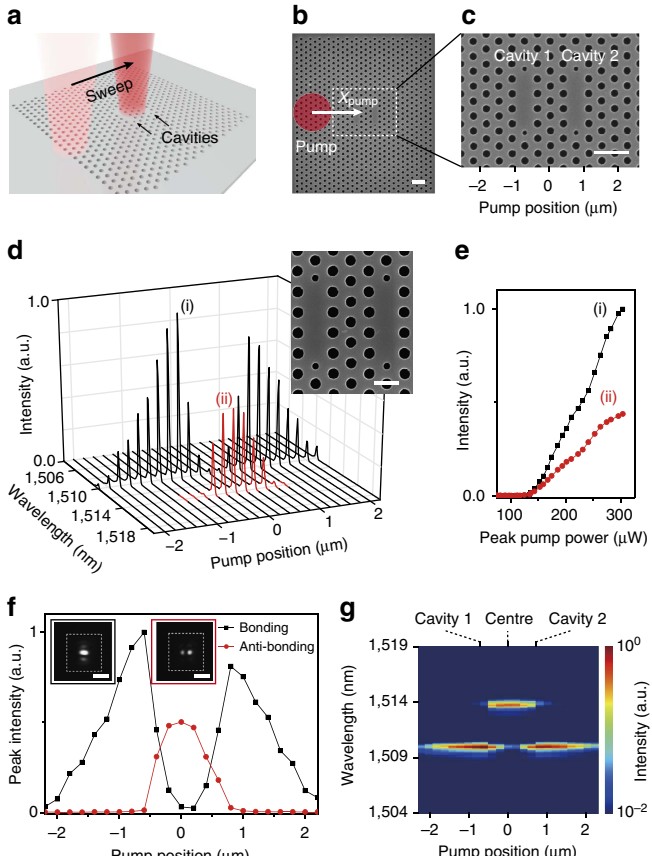

**Figure 2 | Scanning PL measurements in the coupled PhC cavities without graphene.** (**a**) Schematic illustration of scanning PL measurement in coupled cavities without graphene. (**b**) Indication of the pump position ($x_{pump}$, red circle) on the SEM image of fabricated coupled cavities without graphene. A 980-nm pulsed laser diode with a spot size of $\sim 3.0\,\mu m$ was used as a pumping source. The scanning step was $0.2\,\mu m$. Scale bar, $1\,\mu m$. (**c**) Magnified SEM image of panel **b** showing the entire pumping region including cavities 1 and 2. Scale bar, $1\,\mu m$. (**d–g**) Scanning PL measurements in the coupled cavities without graphene as a function of the pump position. The incident peak pump power was $375\,\mu W$. (**d**) Measured PL spectra of the bonding (black lines) and anti-bonding (red lines) modes in the range of pump position from $-2.2$ to $2.2\,\mu m$. Inset, SEM image of the fabricated coupled cavities without graphene. Scale bar, 500 nm. (**e**) L–L curves of the bonding (black line) and anti-bonding (red line) lasing modes. The bonding and anti-bonding modes were optically pumped at $x_{pump} = -0.6$ and $0\,\mu m$, respectively. The lasing threshold of both modes was $\sim 130\,\mu W$. (**f**) Peak intensities of the bonding (black line) and anti-bonding (red line) lasing modes plotted as a function of the pump position. Data were taken from panel **d**. Insets, measured mode images of the bonding (black frame) and anti-bonding (red frame) lasing modes. Scale bars, $5\,\mu m$. (**g**) Log-scale false-colour map of the measured wavelength as a function of the pump position. Data were taken from panel **d**. The peak wavelengths of the bonding and anti-bonding modes were 1510.1 and 1513.8 nm, respectively. The positions of cavities 1 and 2, and the centre of the coupled cavities are indicated.

lasing behaviour. One graphene sheet covers approximately two thirds of the area of cavity 1 (inset of Fig. 3a; large-area graphene), whereas the other covers approximately one third of the area of cavity 1 (inset of Fig. 3f; small-area graphene) (see Methods). All structural parameters between the coupled PhC cavities of Fig. 3a,f were the same. In the PL measurement from the coupled cavities with large-area graphene, only a single

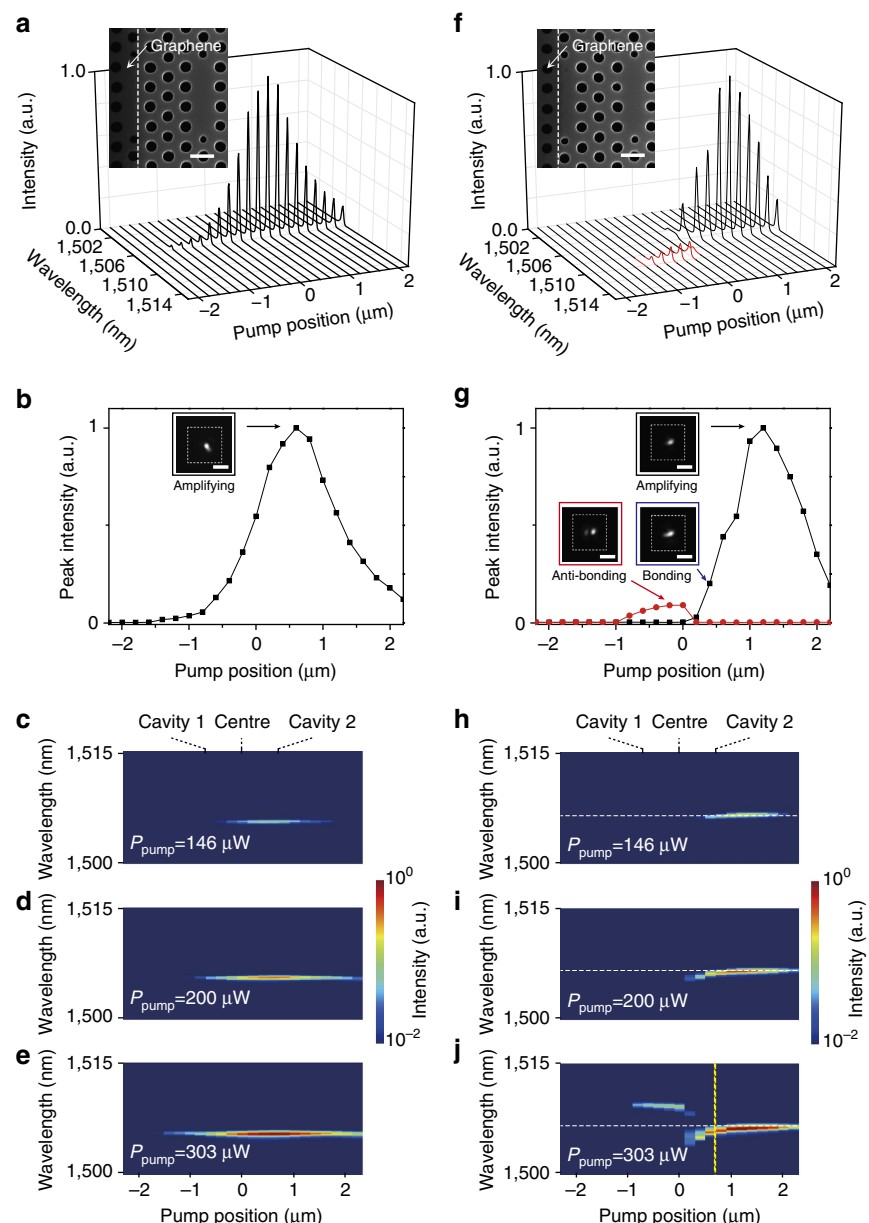

**Figure 3 | Scanning PL measurements in the coupled PhC cavities with graphene.** (**a**–**e**) Measurements in the coupled cavities with large-area graphene. (**a**) PL spectrum versus pump position. The incident peak pump power was 303 μW. The single lasing mode was observed at 1505.6 nm. Inset, SEM image of the fabricated coupled cavities with a graphene layer covering ∼2/3 area of the cavity 1. The boundary of the graphene cover is indicated by a white dashed line. Scale bar, 500 nm. (**b**) Peak intensity of the single lasing mode as a function of the pump position. Data were taken from panel **a**. Inset, measured mode image of the single lasing mode at $x_{pump} = 0.6$ μm, which is strongly confined in cavity 2. Scale bar, 5 μm. (**c**–**e**) Log-scale false-colour maps of the measured wavelengths as a function of the pump position for peak pump powers of 146 (**c**), 200 (**d**) and 303 μW (**e**). (**f**–**j**) Measurements in the coupled cavities with small-area graphene. (**f**) PL spectrum versus pump position. The incident peak pump power was 303 μW. Two lasing peaks were observed at wavelengths of ∼1505 (black lines) and ∼1510 nm (red lines). Inset, SEM image of the fabricated coupled cavities with a graphene layer covering ∼1/3 area of the cavity 1. The structural parameters of the PhC cavities are the same as those in the inset of panel **a**. The boundary of the graphene cover is indicated by a white dashed line. Scale bar, 500 nm. (**g**) Peak intensities of the two lasing modes in panel **f** (black and red lines) as a function of the pump position. Insets, measured mode images of the lasing modes at $x_{pump} = 1.2$ (black frame), 0.4 (blue frame) and −0.2 μm (red frame). Scale bars, 5 μm. (**h**–**j**) Log-scale false-colour maps of the measured wavelengths as a function of the pump position for peak pump powers of 146 (**h**), 200 (**i**) and 303 μW (**j**). The white dashed line at 1506.5 nm was added as a guideline in each panel. We added the yellow dashed line indicating $x_{pump} = 0.6$ μm in panel **j**.

lasing peak was observed at a wavelength of 1505.6 nm for all pump positions (Fig. 3a). The peak intensity of the lasing mode increased monotonically as $x_{pump}$ changed from −2.2 (left end) to 0.6 μm (cavity 2) and decreased as $x_{pump}$ changed from 0.6 (cavity 2) to 2.2 μm (right end). To clearly observe this feature, the lasing peak intensity was plotted as a function of

the pump position (Fig. 3b). The maximum peak intensity was shown at $x_{pump} = 0.6$ μm (cavity 2). Also, a lasing mode image with an intensity anti-node at cavity 2 was captured by an IR camera (inset of Fig. 3b), exhibiting strong light confinement in a smaller region than the bonding mode with a central anti-node (inset of Fig. 2f). In addition, the L–L curves of the

single lasing peak were measured at various pump positions around the position of cavity 2 (Supplementary Fig. 1a). The lasing thresholds were measured to be ∼125 μW, which remained almost unchanged for all pump positions. Furthermore, the log-scale false-colour maps of the measured PL spectra (Fig. 3c–e) show that the peak wavelengths of the single lasing peak were preserved for all pump positions. In particular, no wavelength changes were observed even for different peak pump powers of 146 (Fig. 3c), 200 (Fig. 3d) and 303 μW (Fig. 3e).

The measurements strongly support that the sufficient contrast of asymmetric optical gain was formed in the coupled PhC cavities with large-area graphene. The gain of cavity 1 was significantly suppressed because of the large graphene cover and thus the gain contrast can be larger than the fixed coupling constant ($\Delta\gamma > J$; broken PT-symmetry phase). As a result, no wavelength change of the single lasing mode was observed for different pump positions and powers (Fig. 3c–e and Supplementary Fig. 2a). In addition, the lasing mode was strongly confined only in cavity 2 (inset of Fig. 3b). Furthermore, the lack of change in the lasing threshold at different pump positions (Supplementary Fig. 1a) shows a truly single-mode lasing operation, as a result of the weak interaction between cavities 1 and 2. Therefore, the observed single lasing mode originated from the amplifing mode, which is the optically gaining supermode in the broken PT-symmetry phase (Fig. 1f).

**Measurements in the coupled cavities with small-area graphene.** We repeated scanning PL measurements in the coupled cavities with small-area graphene (Fig. 3f). Two different peaks were observed at wavelengths of ∼1505 and ∼1510 nm, as the pump laser was line-scanned from $x_{pump} = -2.2$ (left end) to 2.2 μm (right end). The lasing peak at ∼1505 nm (black lines) started to appear at $x_{pump} = 0.2$ μm and showed the maximum intensity at $x_{pump} = 1.2$ μm, whereas the peak at ∼1510 nm (red lines) was observed only in the pumping region of $-1.0 < x_{pump} < 0$ μm. This trend was also clearly shown in the plot of peak intensities as a function of the pump position (Fig. 3g). The lasing peak at ∼1505 nm (black line) exhibits much higher peak intensities than the peak at ∼1510 nm (red line). In addition, we observed three different lasing mode images at $x_{pump} = -0.2$, 0.4 and 1.2 μm (insets of Fig. 3g). Interestingly, the anti-bonding mode with a central intensity node, the bonding mode with a central anti-node and the amplifing mode with an intensity anti-node at cavity 2 were clearly visible at $x_{pump} = -0.2$, 0.4 and 1.2 μm, respectively. The position-dependent excitation of these different lasing modes was also shown in the measurements of L–L curves as a function of the pump position (Supplementary Fig. 1b). Noticeable changes of the lasing thresholds were observed as $x_{pump}$ changed by approximately ±0.2 μm from the positions of 0 and 0.6 μm, respectively, which indicate the transitions of lasing modes at $x_{pump} \sim 0$ and 0.6 μm. Furthermore, we note that the peak wavelengths of these lasing modes vary with changing pump position (Fig. 3f), in contrast to the results of Figs 2d and 3a. To further study the behaviour of such a wavelength shift, we examined different peak pump powers of 146, 200 and 303 μW, and plotted the log-scale false-colour maps of these measured PL spectra (Fig. 3h–j). The PL maps show significant pump power dependence, including following features. First, no wavelength change in the single-mode lasing near 1505 nm was observed for the peak pump power of 146 μW. Second, a red-shift of the lasing mode occurred only in the range of $0 < x_{pump} < 0.6$ μm for the peak pump powers higher than 200 μW, whereas the lasing wavelength remained constant at $x_{pump} > 0.6$ μm. A further red-shift was observed at $0 < x_{pump} < 0.6$ μm with increasing peak pump power and, subsequently, the lasing wavelength changed

from 1504.5 to 1506.5 nm as $x_{pump}$ changed from 0 to 0.6 μm at 303 μW. Third, the lasing peak near 1510 nm was not observed when the peak pump power was lower than 303 μW. At 303 μW, however, the lasing wavelength was slightly blue-shifted from 1509.8 to 1509.3 nm as $x_{pump}$ changed from −1.0 to 0 μm. Taken together, distinctive lasing behaviour was clearly observed in three pumping regions: $-1.0 < x_{pump} < 0$ μm, $0 < x_{pump} < 0.6$ μm and $x_{pump} > 0.6$ μm.

The unique optical features measured in the coupled PhC cavities with small-area graphene were attributed to an efficient gain control through tuning of the area of the graphene cover as well as the pump position and power. In particular, the transition of lasing modes at $x_{pump} \sim 0.6$ μm can be well understood by the asymmetric gain contrast applied in the experiment. First, sufficient gain was not provided to cavity 1 at the low peak pump power of 146 μW (Fig. 3h), although the graphene area on cavity 1 was reduced compared with that of the coupled cavities of Fig. 3a and the region near cavity 1 was efficiently optically pumped. Therefore, the gain contrast between cavities 1 and 2 was still relatively large in this case and no wavelength change was observed in the single lasing mode, similarly to the coupled cavities with large-area graphene in Fig. 3a–e ($\Delta\gamma > J$; broken PT-symmetry phase). As the peak pump power increased up to 303 μW (Fig. 3j; Supplementary Fig. 2b), the effective optical gain of cavity 1 became large enough to excite the two lasing modes, bonding (∼1505 nm) and anti-bonding (∼1510 nm) modes, by optically pumping the region near cavity 1 ($-1.0 < x_{pump} < 0.6$ μm). This exhibits smaller gain contrast that is similar to the observation in the coupled cavities without graphene in Fig. 2d–g ($\Delta\gamma < J$; unbroken PT-symmetry phase). On the other hand, when the pump laser was in the region of cavity 2 ($0.6 < x_{pump} < 2.2$ μm), the single-mode lasing with a constant peak wavelength and the amplifing mode image was observed again because of the reduced optical gain of cavity 1 ($\Delta\gamma > J$). Consequently, the direct phase transition occurred from the bonding and anti-bonding modes ($x_{pump} < 0.6$ μm) to the single-amplifing mode ($x_{pump} > 0.6$ μm) at the pump power of 303 μW. The position of $x_{pump} = 0.6$ μm is the visualized EP. This phase transition is also strongly supported by the measured images of three lasing modes (insets of Fig. 3g) and the threshold change at $x_{pump} \sim 0.6$ μm (Supplementary Fig. 1b). In fact, the bonding mode ($0 < x_{pump} < 0.6$ μm) exhibits identical field profiles in cavities 1 and 2 and experiences additional optical loss owing to the graphene on cavity 1. Therefore, the lasing threshold of the bonding mode became larger than that of the amplifing mode. In addition, the lasing behaviours near $x_{pump} \sim 0$ μm can be explained by mode competition: only one lasing mode between the bonding and anti-bonding modes was observed at $-1.0 < x_{pump} < 0.6$ μm, similarly to Fig. 2f,g (ref. 39).

**Identification of lasing modes.** To fully understand our experimental results, we investigated the measured PL spectra as a function of the pump position in the coupled cavities with large-area/small-area graphene covers and without graphene, using the theoretical model of non-Hermitian coupled optical cavities with asymmetric gains in Equation 1 (Fig. 4). As the focused pump light with a spot size of ∼3.0 μm sweeps the coupled cavities in the experiment, the induced optical gains of cavities 1 and 2 experience spatial distributions that vary as a function of the pump position. In our model, we reflect this situation by setting the optical gains of cavities 1 and 2 as simplified Gaussian gain profiles with a full-width at half-maximum of 3.0 μm and central positions of $x_{pump} = -0.6$ and 0.6 μm, respectively. The gain of cavity 1 also needs to be

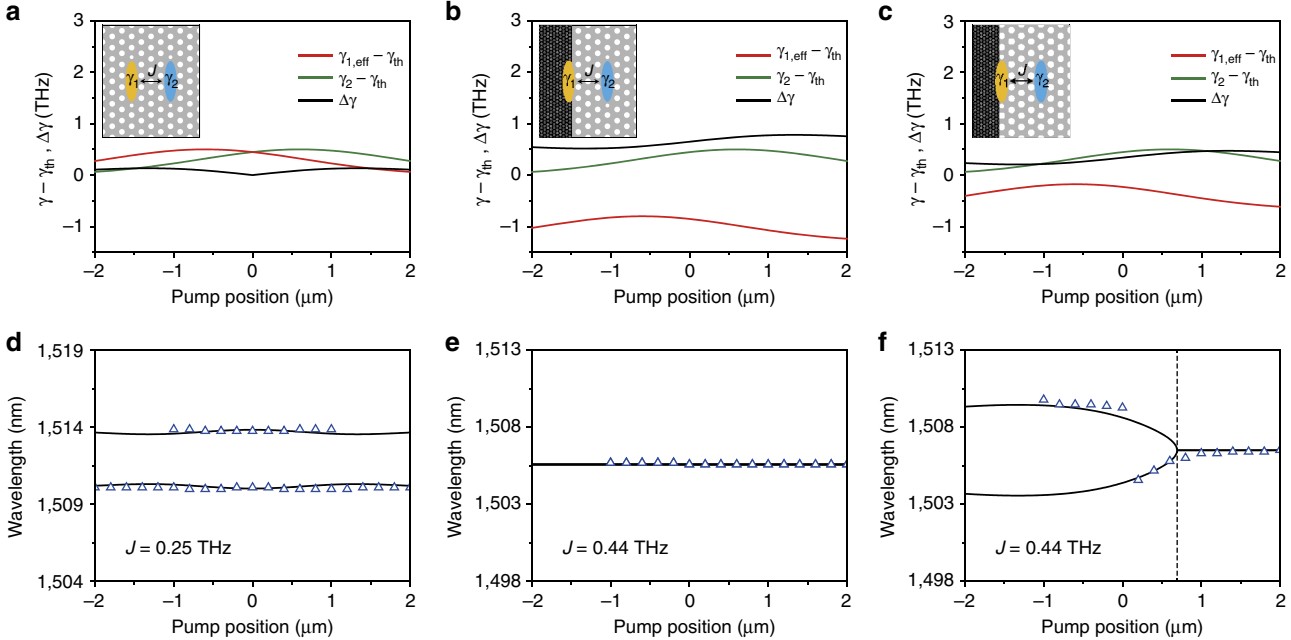

**Figure 4 | Identification of lasing modes using asymmetric gain model.** (**a–c**) Spatial profiles of the net optical gains of cavity 1 and cavity 2, $\gamma_{1,\mathrm{eff}} - \gamma_{\mathrm{th}}$ (red lines) and $\gamma_2 - \gamma_{\mathrm{th}}$ (green lines), in the coupled PhC cavities without graphene (**a**), with large-area graphene (**b**) and with small-area graphene (**c**). The effective gain of cavity 1 was set to $\gamma_{1,\mathrm{eff}} = \gamma_1 - \kappa_{\mathrm{graphene}}$, where $\kappa_{\mathrm{graphene}}$ is the optical loss of graphene. $\gamma_{\mathrm{th}}$ is the threshold gain for the lasing of a single PhC cavity, which is the same as $\kappa$. Gaussian profiles with a full-width at half-maximum of 3.0 µm and the central positions of $x_{\mathrm{pump}} = -0.6$ and 0.6 µm were employed to represent the net optical gains of cavities 1 and 2, respectively. Different values of $\kappa_{\mathrm{graphene}}$ were examined based on the area of graphene cover: $\kappa_{\mathrm{graphene}} = 0$ (**a**), 1.2 (**b**) and 0.68 THz (**c**). The gain contrast between the net gains is $\Delta\gamma = 1/2 \times |\gamma_2 - \gamma_{1,\mathrm{eff}}|$ (black lines). Insets, schematic illustrations of the coupled cavities without and with graphene. (**d–f**) Calculated resonance wavelengths, $c/\mathrm{Re}(f)$, where $c$ is the speed of light, were plotted as a function of the pump position (black solid lines) in the coupled PhC cavities without graphene (**d**), with large-area graphene (**e**) and with small-area graphene (**f**). Measured peak wavelengths were plotted together (triangles): the measured data in panels **d**,**e** and **f** are taken from Figs 2g and 3e,j, respectively. Experimentally determined wavelengths and coupling constants were used in the calculations: $(c/f_0, J) = (1512.0\ \mathrm{nm}, 0.25\ \mathrm{THz})$ (**d**), $(1505.6\ \mathrm{nm}, 0.44\ \mathrm{THz})$ (**e**) and $(1506.5\ \mathrm{nm}, 0.44\ \mathrm{THz})$ (**f**). The EP at $x_{\mathrm{pump}} = 0.7$ µm is indicated by a black dashed line in panel **f**.

considered as an effective gain, $\gamma_{1,\mathrm{eff}} = \gamma_1 - \kappa_{\mathrm{graphene}}$, where $\kappa_{\mathrm{graphene}}$ is the optical loss of graphene, while the gain of cavity 2 is not affected by graphene (see Methods). Moreover, as we observed lasing modes in the entire pumping range from $-2.2$ to $2.2$ µm, it is reasonable to define the net optical gains of cavities 1 and 2 as $\gamma_{1,\mathrm{eff}} - \gamma_{\mathrm{th}}$ (red lines in Fig. 4a–c) and $\gamma_2 - \gamma_{\mathrm{th}}$ (green lines in Fig. 4a–c), respectively, where $\gamma_{\mathrm{th}}$ is the threshold gain for the lasing of a single PhC cavity. The gain contrast between the net gains was calculated as $\Delta\gamma = 1/2 \times |\gamma_2 - \gamma_{1,\mathrm{eff}}|$ (black lines in Fig. 4a–c). Using the experimentally determined $f_0$ and $J$ in each coupled cavity, we calculated $\mathrm{Re}(f)$ from Equation 1 and the corresponding resonance wavelength, $c/\mathrm{Re}(f)$, where $c$ is the speed of light. The value of $J$ was obtained from the wavelength difference between the bonding and anti-bonding modes in Figs 1i and 2d, and also confirmed by numerical simulations (see Methods).

First, in the coupled PhC cavities without graphene of Fig. 2d–g (Fig. 4a), the resonance wavelengths were calculated using $c/f_0 = 1512.0$ nm, $J = 0.25$ THz, and $\kappa_{\mathrm{graphene}} = 0$, and plotted as a function of the pump position (black solid lines, Fig. 4d) with the measured peak wavelengths of the bonding and anti-bonding lasing modes in Fig. 2g (triangles, Fig. 4d). Both the calculated and measured resonance wavelengths are almost independent from the pump position. Second, in the coupled cavities with large-area graphene cover of Fig. 3a–e, the resonance wavelengths were calculated with Equation 1 using $c/f_0 = 1505.6$ nm, $J = 0.44$ THz and $\kappa_{\mathrm{graphene}} = 1.2$ THz. The constant wavelengths with varying pump position (black solid line, Fig. 4e) agree well with the measured peak wavelengths

of the amplifying lasing mode in Fig. 3e (triangles, Fig. 4e). Third, in the coupled cavities with small-area graphene cover of Fig. 3f–j, we assumed $\kappa_{\mathrm{graphene}} = 0.68$ THz, based on the comparison of the area of graphene cover with that in the coupled cavities with large-area graphene (see Methods). The resonance wavelengths calculated using $c/f_0 = 1506.5$ nm and $J = 0.44$ THz (black solid lines, Fig. 4f) reproduced well the measured peak wavelengths in Fig. 3j (triangles, Fig. 4f), showing the transition from the bonding/anti-bonding lasing modes (unbroken PT-symmetry; $x_{\mathrm{pump}} < 0.7$ µm) to the single-amplifying lasing mode (broken PT-symmetry; $x_{\mathrm{pump}} > 0.7$ µm). The EP at $x_{\mathrm{pump}} = 0.7$ µm (black dashed line, Fig. 4f) is almost same as the one observed in the experiment of Fig. 3j. Furthermore, we calculated $\mathrm{Im}(f)$ from Equation 1 with varying pump positions (Supplementary Fig. 3) to elucidate the appearance of the lasing modes in the coupled cavities with and without graphene. As expected in Fig. 1c, the measured lasing peaks emerged only in the pump positions with $\mathrm{Im}(f) > 0$.

**Tuning of the exceptional points.** Finally, we successfully demonstrated active tuning of the EPs by controlling the optical loss of graphene, $\kappa_{\mathrm{graphene}}$, through electrical gating[36–38,40]. Ion gel was placed on the coupled PhC cavities with partially covered graphene (see Methods; Supplementary Fig. 4a). Scanning PL measurements were performed, while a gate voltage $V_g$ was applied to the graphene with ion gel (Fig. 5a). The pump laser was line-scanned from $-2.0$ to $2.0$ µm with a peak pump power of 843 µW. Then, the wavelengths of the measured resonant peaks

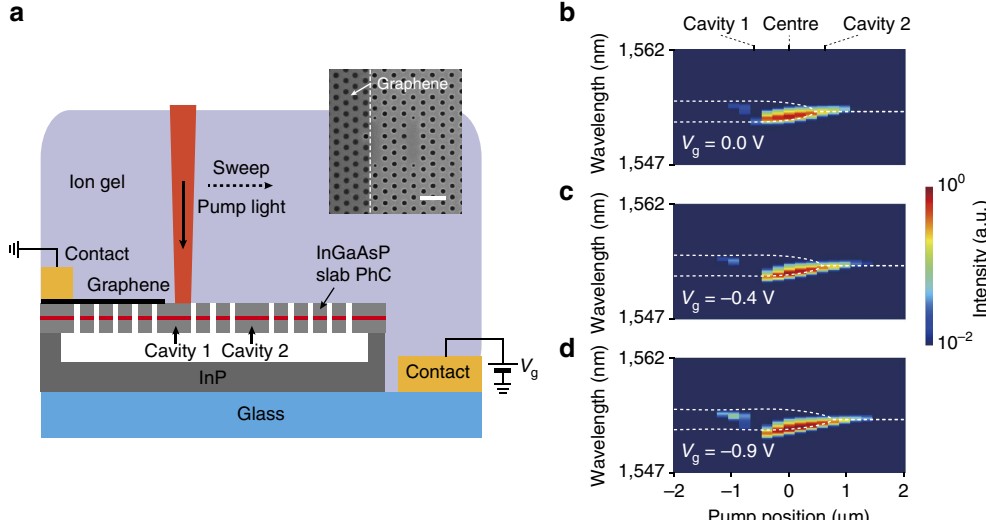

**Figure 5 | Tunable EPs.** (**a**) Schematic illustration of coupled PhC cavities with graphene. Scanning PL measurements were performed while a gate voltage $V_g$ was applied to the graphene with ion gel on top. Inset, SEM image of the fabricated coupled cavities with a monolayer graphene sheet before introducing ion gel. The boundary of graphene is indicated by a white dashed line. Scale bar, 1 µm. (**b**–**d**) Log-scale false-colour maps of the measured PL spectra as a function of pump position, at $V_g = 0.0$ (**b**), $-0.4$ (**c**) and $-0.9$ V (**d**). The incident peak power of the pump laser was 843 µW. The calculated resonance wavelengths were plotted as a function of pump position (white dashed lines). EPs were observed at $x_{pump} = 0.45$, 0.57 and 0.75 µm for $V_g = 0.0$, $-0.4$ and $-0.9$ V, respectively.

changed at different gate voltages (Fig. 5b–d). To estimate EPs at different gate voltages, using the procedure of Fig. 4, we also measured light transmission in a graphene/ion gel structure without PhC cavities with varying $V_g$ from 0 to $-1.1$ V, and determined $V_g$-dependent graphene loss, $\kappa_{graphene}$ (Supplementary Fig. 4b). The value of $\kappa_{graphene}$ was set to 0.20, 0.17 and 0.13 THz for $V_g = 0.0$, $-0.4$ and $-0.9$ V, respectively. In addition, in the coupled PhC cavities with graphene, experimentally determined $c/f_0$ and $J$ were 1554 nm and 0.17 THz, respectively, for all gate voltages. Then, EPs were observed at $x_{pump} = 0.45$, 0.57 and 0.75 µm for $V_g = 0.0$, $-0.4$ and $-0.9$ V, respectively (white dashed lines, Fig. 5b–d). As $|V_g|$ increased or optical loss of graphene was reduced, the pump position for EP became larger as expected from Fig. 4. We note that this is an explicit experimental demonstration of tunable EPs through electrical gating.

## Discussion

In summary, we have demonstrated the phase transition of lasing modes and observed EPs in the coupled PhC cavities with graphene—the non-Hermitian wavelength-scale optical system. Sufficient contrast of asymmetric optical gain was provided by placing a monolayer graphene sheet on top of one cavity and tuning the area of graphene cover. The control of gain contrast by graphene is useful particularly in the wavelength-scale coupled cavities because conventional optical pumping methods are not efficient to provide asymmetric gain as the two cavities are closely located. Furthermore, the optical loss of graphene was tuned by electrical gating with ion gel, and as a result, active tuning of the EP was successfully demonstrated. We believe that our coupled PhC cavities with graphene are useful as a powerful platform to investigate unique features of non-Hermitian systems and demonstrate new PT-symmetric optical devices. Various practical applications such as the implementation of tunable EPs, the excitation of desired lasing modes and the demonstration of efficient all-optical switching are feasible in our system, by controlling the optical loss of graphene.

## Methods

**Numerical simulations.** Full-wave simulations using finite element methods (FEM; COMSOL Multiphysics, wave optics module) were performed to calculate the complex eigenfrequencies of supermodes (Fig. 1b,c) and the corresponding mode profiles (Fig. 1d–g) in the coupled three-cell PhC cavities. The FEM solved the three-dimensional Helmholtz equation,

$$\nabla \times \nabla \times \mathbf{E}(\mathbf{x}) - k_0^2 \varepsilon_r(\mathbf{x}) \mathbf{E}(\mathbf{x}) = 0 \qquad (2)$$

with appropriate structural parameters and boundary conditions. We applied scattering boundary conditions at all boundaries of the calculation domain to reduce the size effect of the finite domain. In Fig. 1b–g, the structural parameters of the coupled PhC cavities were obtained based on the SEM image of the fabricated structure in Fig. 1i: the lattice constant was 420 nm, regular hole diameter was 265 nm, the reduced hole diameter at the cavity boundary was 140 nm (these air holes were moved outward 63 nm from the regular positions), and slab thickness was 250 nm. The refractive index of the InGaAsP slab, $\mathrm{Re}(n) + \mathrm{iIm}(n)$, was set to $3.3 + \mathrm{i}0.01$ (ref. 41). To provide the gain or loss in the cavities, we changed $\mathrm{Im}(n)$ only inside each cavity with a size of $1.46 \times 2.52 \times 0.25$ µm³. The simulations were then performed varying $\mathrm{Im}(n)$ from 0.01 to $-0.01$ in each cavity separately, which correspond to the optical gains varying from 0 to 1.106 THz. We considered only the fundamental mode in each single PhC cavity. In addition, the coupling constants $J$ were calculated in Fig. 4d–f based on the SEM images of the fabricated structures (insets of Figs 2d and 1i). The diameters of the central nine holes between the two cavities in the $\Gamma$–$K$ direction were 0.9 times reduced (Fig. 4d) or 1.1 times enlarged[42] (Fig. 4e,f).

**Device fabrication.** PhC structures were fabricated on a 250-nm-thick InGaAsP/800-nm-thick InP/100-nm-thick InGaAs/InP substrate wafer using electron-beam lithography and chemically assisted ion-beam etching. The InGaAsP slab included a single quantum well with a central emission wavelength of ∼1.5 µm. Two identical three-cell PhC cavities with a lattice constant of 420 nm were formed in the slab. The sacrificial InP layer underneath the InGaAsP slab was selectively wet etched using a diluted HCl:H₂O (3:1) solution. Next, chemical vapour deposition-grown monolayer graphene was coated with poly(methyl methacrylate) and transferred to the fabricated PhC samples. Additional aligned electron-beam lithography was carried out to define a poly(methyl methacrylate) mask to protect the graphene layer on a desired position. By the subsequent oxygen plasma etching process, the monolayer graphene was placed on top of only one PhC cavity. The graphene cover areas on cavity 1 were measured as $0.62 \times 2.52$ µm² and $0.35 \times 2.52$ µm² in the coupled cavities with large-area graphene (Fig. 3a) and small-area graphene (Fig. 3f), respectively. In Fig. 5, the entire sample including the graphene-PhC structures was attached to a glass substrate with two large Ti/Au contacts. Ion gel was cut with a razor blade, and transferred onto the graphene-PhC structures using tweezers.

**Optical measurement.** A 980-nm pulsed laser diode (10-ns pulses with 1% duty cycle) was used to optically pump the coupled PhC cavities at room temperature. The light emitted from the cavities was collected by a ×40 microscope objective lens with a numerical aperture of 0.55 and focused onto either an IR 1D array detector (PyLoN, Princeton Instruments) or an InGaAs IR camera (C10633, Hamamatsu). To control the pump position in the scanning PL measurement, the motorized single-axis translation stage was used to move the PhC samples with a scanning step of 0.2 μm, while a fixed pump laser with a spot size of ~3.0 μm illuminated the samples. The pump positions were accurately determined using a reference position near the coupled PhC cavities. In Fig. 5, a gate voltage was applied with ion gel using a DC voltage source (R6142, Advantest).

**Data availability.** The data that support the findings of this study are available from the corresponding author on request.

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

## Acknowledgements

We would like to thank Prof W.I. Park for providing monolayer graphene and Prof. S.-H. Kwon for helpful discussions. This work was supported by the National Research Foundation of Korea (NRF) grant funded by the Korean government (MSIP) (nos. 2009-0081565 and 2014M3A6B3063710).

## Author contributions

K.-H.K. and H.-G.P. designed the experiments. K.-H.K., M.-S.H., H.-R.K. and J.-H.C. performed the experiments. K.-H.K. performed the simulations and Y.-S.N. analysed the data. K.-H.K. and H.-G.P. wrote the manuscript. All authors discussed the results and commented on the manuscript.

## Additional information

**Competing financial interests:** The authors declare no competing financial interests.

