## [Peer review file · Nature Communications]

Reviewer #1 (Remarks to the Author):

The work by K.H Kim investigates experimentally PT symmetric lasers in coupled photonic crystal cavities. In this work, they demonstrate lasing in the PT broken phase and in the PT symmetric phase. The authors controlled the gain by two different methods: (1) by covering one of the cavities by a layer of graphene, and (2) by scanning the location of the optical pump. In addition, they used a combination of both methods to further confirm their results.

Furthermore, the authors used a combination of full wave analysis and a theoretical model based coupled mode theory in order to validate their experimental results.

In my opinion, this is the most clean experimental result on PT symmetric lasers and the paper is by far the most clearly written one with all the experimental and theoretical parameters presented in details. In summary, the presented work is of high quality and will have high impact on the field.

I definitely recommend this work for publications in Nature Communications after the authors consider the following minor comments:

1. It appears that, in the absence of graphene layers, the anti-bonding mode dominates the emission when the pump beam is centered. Is there a simple explanation for this observation?
2. Does introducing a graphene layer shifts the cavity frequency?
3. In the sentence: "Scanning photoluminescence (PL) measurements show the bonding/anti-bond lasing modes," The authors use "anti-bond" which is inconsistent with the rest of the manuscript where they always used: "anti-bonding". I would suggest to make it all the same.
4. There are couple of theoretical papers that also propose to use the properties of exceptional points in multimode laser arrays to engineer the emission. The authors might find them useful: Phys Rev. A 92, 033818 (2015) and arXiv:1608.04618

Reviewer #3 (Remarks to the Author):

MS by K. Kim, et al., with the title "Direct observation of exceptional points in coupled photonic-crystal lasers with asymmetric optical gains" is considering two coupled laser cavity where one of the cavities is partially covered by graphene. The graphene helps to have absorption in one of the cavities and as a result the pump will generate a non-symmetric gain profile in the cavities. Authors are claiming the observation of Exceptional point in the cavities. A transition from double mode lasing to single mode lasing has been depicted experimentally.

I admit that the paper is well written. Some citations are missing, among them are the following:

- 1- Journal of Physics A: Mathematical and Theoretical 45 (44), 444029 (2012) By J. Schindler, et al.
- 2- Nature communications 7, (2016) By C. Shi, et al.
- 3- Nature 488, 167–171 (09 August 2012) doi:10.1038/nature11298
- 4- Physical Review A 89 (4), 043842 (2014) By M. Chitsazi, et al.

My main concern is that the results of the paper are not new. All the results have been already demonstrated in other setups. There are many experimental observation of EP point in the literature. Several of them are already have been cited in the current MS. For example, see citation 3, 5, 13. I do not stress again the citations that I was proposing.

In this respect I am not sure why one wants to stress what has been already demonstrated. Does the setup that authors proposing has any advantage with respect to all other proposals? There should be a strong reason that the proposed structure provides to the community. Otherwise I am not sure if the Nat. Com. can accept a paper that demonstrate a redundant result.

- Response to Reviewer 1.

Comment. The work by K.H Kim investigates experimentally PT symmetric lasers in coupled photonic crystal cavities. In this work, they demonstrate lasing in the PT broken phase and in the PT symmetric phase. The authors controlled the gain by two different methods: (1) by covering one of the cavities by a layer of graphene, and (2) by scanning the location of the optical pump. In addition, they used a combination of both methods to further confirm their results.

Furthermore, the authors used a combination of full wave analysis and a theoretical model based coupled mode theory in order to validate their experimental results.

In my opinion, this is the most clean experimental result on PT symmetric lasers and the paper is by far the most clearly written one with all the experimental and theoretical parameters presented in details. In summary, the presented work is of high quality and will have high impact on the field.

I definitely recommend this work for publications in Nature Communications after the authors consider the following minor comments:

Author Response. We thank the reviewer for the strongly positive comments about the novelty and importance of our PT symmetric lasers in coupled photonic crystal cavities.

Comment 1. It appears that, in the absence of graphene layers, the anti-bonding mode dominates the emission when the pump beam is centered. Is there a simple explanation for this observation?

Author Response. The anti-bonding mode is confined mostly in cavity 1 and cavity 2, whereas the bonding mode has significant electric fields between the cavities as well as in the cavities (Fig. 1d). Thus, the anti-bonding mode can dominate the emission when both cavity 1 and cavity 2 are pumped simultaneously or the pump beam is centered. This was also observed in ref. 29. We have added a simple explanation for this observation, as follows (line 13 on page 7): “the different field overlap with the pump laser²⁹ and”.

Comment 2. Does introducing a graphene layer shifts the cavity frequency?

Author Response. No. As we already mentioned in the manuscript (line 1 on page 6), introducing a graphene layer shows a negligibly small change in the intrinsic eigenfrequency of a single cavity mode (refs. 35-38 in the new submission).

Comment 3. In the sentence: “Scanning photoluminescence (PL) measurements show the bonding/anti-bond lasing modes, ...” The authors use “anti-bond” which is inconsistent with the rest of the manuscript where they always used: “anti-bonding”. I would suggest to make it all the same.

Author Response. We appreciate the reviewer checking the word. We changed ‘anti-bond’ into ‘anti-bonding’ in the manuscript.

Comment 4. There are couple of theoretical papers that also propose to use the properties of exceptional points in multimode laser arrays to engineer the emission. The authors might find them useful: Phys Rev. A 92, 033818 (2015) and arXiv:1608.04618

Author Response. We have included these references (refs. 11 and 14 in the new submission) in our manuscript.

- Response to Reviewer 2.

Comment. MS by K. Kim, et al., with the title “Direct observation of exceptional points in coupled photonic-crystal lasers with asymmetric optical gains” is considering two coupled laser cavity where one of the cavities is partially covered by graphene. The graphene helps to have absorption in one of the cavities and as a result the pump will generate a non-symmetric gain profile in the cavities. Authors are claiming the observation of Exceptional point in the cavities. A transition from double mode lasing to single mode lasing has been depicted experimentally.

I admit that the paper is well written. Some citations are missing, among them are the following:

1- Journal of Physics A: Mathematical and Theoretical 45 (44), 444029 (2012) By J. Schindler, et al.

2- Nature communications 7, (2016) By C. Shi, et al.

3- Nature 488, 167–171 (09 August 2012) doi:10.1038/nature11298

4- Physical Review A 89 (4), 043842 (2014) By M. Chitsazi, et al.

Author Response. We thank the reviewer for the assessment of our work. We have included these references (refs. 6, 7, 9 and 13 in the new submission) in our manuscript.

Comment. My main concern is that the results of the paper are not new. All the results have been already demonstrated in other setups. There are many experimental observation of EP point in the literature. Several of them are already have been cited in the current MS. For example, see citation 3, 5, 13. I do not stress again the citations that I was proposing.

In this respect I am not sure why one wants to stress what has been already demonstrated. Does the setup that authors proposing has any advantage with respect to all other proposals? There should be a strong reason that the proposed structure provides to the community. Otherwise I am not sure if the Nat. Com. can accept a paper that demonstrate a redundant result.

Author Response. We respectfully but strongly disagree with the reviewer. Exceptional points (EPs) have been observed in various physical and optical systems, as the reviewer pointed out. However, in this work, we present the first demonstration of parity-time (PT) symmetric lasers in *wavelength-scale* coupled photonic-crystal cavities with a *monolayer graphene sheet*. The monolayer graphene partially covered only one cavity to provide asymmetric optical gain and sufficient gain contrast, as well as to enable active *tuning* of gain contrast by electrical gating. Indeed, *we successfully demonstrated active tuning of EPs by controlling the optical loss of graphene, for the first time* (Fig. 5). In addition, we used coupled photonic-crystal cavities with submicrometer-sized footprints, and thus, only few lasing modes were supported, clear mode identification and accurate analysis of lasing behaviors were performed, and intercavity coupling between two identical cavities was precisely controlled. Therefore, we strongly believe that our approach is a powerful platform to undoubtedly elucidate unique features of the wavelength-scale non-Hermitian optical systems and achieve their practical implementations. As the reviewer 1 mentioned, we also believe that this work is the most clean experimental result on PT symmetric lasers, and is of high quality and will have high impact on the field.

To emphasize the role of the graphene layer and the distinction from previous work on PT symmetric lasers, we added two more sentences to the revised manuscript as follows (line 14 on page 14): “The control of gain contrast by graphene is useful particularly in the wavelength-scale coupled cavities because conventional optical pumping methods are not efficient to provide asymmetric gain as the two cavities are closely located. Furthermore, the optical loss of graphene was tuned by electrical gating with ion gel, and as a result, active tuning of the EP was successfully demonstrated.” In addition, to elaborate on the tuning of exceptional point, we added several sentences to the section of ‘Tuning of the exceptional points’ of the manuscript (line 22 on page 13): “The pump laser was line-scanned from -2.0 to 2.0 μm with a peak pump power of 843 μW .” and “To estimate EPs at different V_g ’s, we also measured light transmission in a graphene/ion gel structure without PhC cavities with varying V_g from 0 to -1.1 V, and determined V_g -dependent graphene loss, κ_{graphene} (Supplementary Fig. 4b). κ_{graphene} were set to 0.20 , 0.17 and 0.13 THz for $V_g = 0.0$, -0.4 and -0.9 V, respectively. Experimentally determined c/f_0 and J were 1554 nm and 0.17 THz, respectively, for all V_g ’s. Then, ”. These sentences were in the caption of Fig. 5 in the original submission. By moving them into the main text, we want to explain the tuning of exceptional point in more detail. Also, during the revision, the value of κ_{graphene} at $V_g = -0.4$ V was slightly changed from 0.16 to 0.17 for better analysis, and as a result, the exceptional point at $V_g = -0.4$ V was also slightly changed from 0.61 to 0.57 . Furthermore, we have included an image of the fabricated device of Fig. 5 and the measurement of tuning of graphene loss in the new Supplementary Fig. 4.

REVIEWERS' COMMENTS:

Reviewer #1 (Remarks to the Author):

The authors has addressed all my comments successfully. I definitely recommend this work for publication in Nature Communications. Upon checking Ref. 14 of the manuscript, I find that it is now published (Scientific Reports 6, 33253 (2016)) and the citation should be thus updated.

Reviewer #3 (Remarks to the Author):

I read the reply very carefully. As I said previously, the MS is well written and the experiment is done in a clear manner. I think the authors clarify my concern in the revised MS and in their reply. The use of graphene as a controllable absorber is now clearly stated where helps to tune the EP point.

- Response to Reviewer 1.

Comment. The authors has addressed all my comments successfully. I definitely recommend this work for publication in Nature Communications. Upon checking Ref. 14 of the manuscript, I find that it is now published (Scientific Reports 6, 33253 (2016)) and the citation should be thus updated.

Author Response. We thank the reviewer for the positive assessment of our work. The reference 14 was updated in the revised manuscript.

- Response to Reviewer 3.

Comment. I read the reply very carefully. As I said previously, the MS is well written and the experiment is done in a clear manner. I think the authors clarify my concern in the revised MS and in their reply. The use of graphene as a controllable absorber is now clearly stated where helps to tune the EP point.

Author Response. We thank the reviewer for the thoughtful comment and positive assessment of our work.